# The Impact of Nurse Health-Coaching Strategies on Cognitive—Behavioral Outcomes in Older Adults

**DOI:** 10.3390/ijerph20010416

**Published:** 2022-12-27

**Authors:** Kathleen Potempa, Susan Butterworth, Marna Flaherty-Robb, Margaret Calarco, Deanna Marriott, Bidisha Ghosh, Amanda Gabarda, Jordan Windsor, Stacia Potempa, Candia Laughlin, Karen Harden, Patricia Schmidt, Alexis Ellis, Philip Furspan

**Affiliations:** 1School of Nursing, University of Michigan, Ann Arbor, MI 48109, USA; 2School of Medicine, University of Tennessee Health Science Center, Memphis, TN 38163, USA; 3Q Consult, St. Petersburg, FL 33707, USA

**Keywords:** nurse health coaching, social determinants of health, change talk, health behavior change, natural language analysis

## Abstract

The practice of nurse health coaching (NHC) draws from the art and science of nursing, behavioral sciences, and evidence-based health-coaching methods. This secondary analysis of the audio-recorded natural language of participants during NHC sessions of our recent 8-week RCT evaluates improvement over time in cognitive–behavioral outcomes: change talk, resiliency, self-efficacy/independent agency, insight and pattern recognition, and building towards sustainability. We developed a measurement tool for coding, Indicators of Health Behavior Change (IHBC), that was designed to allow trained health-coach experts to assess the presence and frequency of the indicators in the natural language content of participants. We used a two-step method for randomly selecting the 20 min audio-recorded session that was analyzed at each time point. Fifty-six participants had high-quality audio recordings of the NHC sessions. Twelve participants were placed in the social determinants of health (SDH) group based on the following: low income (<USD 20,000/year), early-onset hypertension, and social disadvantages. Our analyses significantly improved change talk and the other four factors over time. Our factor analyses indicated two distinct factors at each measurement point of the study, demonstrating the stability of the outcome measures over time. Our newly developed measurement tool, IHBC, proved stable in structure over time and sensitive to change. This NHC program shows promise in improving cognitive–behavioral indicators associated with health behavior change in both non-SDH and SDH individuals.

## 1. Introduction and Background

Nurse health coaching (NHC) has been a recognized practice area of nursing for years [1], with identified and formalized standards and competencies since 2013 [2,3]. We recently completed a randomized trial (RCT) of an NHC model in participants 50 years of age and older with one or more chronic conditions—the protocol and primary outcomes of which are published elsewhere [4,5]. One of the major findings of the RCT was the significant improvement in health habits, including choice of foods, use of alcohol, and activity/exercise levels of the participants of the NHC 8-week intervention group compared to those of the control participants. Past RCTs have shown mixed results of health coaching on behavioral risk outcomes [6], leading to the current emphasis on health behavior science to elucidate the specific mechanism(s) of successful behavior change [1,4].

Coaching strategies related to cognitive/behavioral outcomes were the focus of NHC in the intervention trial. This secondary analysis of the audio-recorded natural language of participants during NHC sessions of the RCT aimed to evaluate improvement in cognitive–behavioral outcomes. Our purpose here is to report on the results of this analysis.

Theoretical Framework. Figure 1 illustrates our nurse health-coaching framework with six strategies: evoking narratives, goal setting, guiding insight/pattern recognition, cognitive restructuring, motivational interviewing, and encouraging sustainability/resiliency. These strategies aim to foster cognitive/behavioral outcomes, which we postulate are the driving forces or mechanisms of achieving health/risk goals. While we propose that the driving forces are distinct factors that are identifiable in the natural language of people, they are highly integrated in that improvement in one factor influences other factors. For example, resiliency is a distinctly recognizable factor that also influences sustainability (see factor descriptions below). The strategies also address restraining forces such as sustain talk and low self-esteem, which impede progress toward goal achievement. Additionally, our NHC framework considers the socio-economic and environmental factors affecting health—the social determinants of health. The cognitive/behavioral outcomes are described below, including examples of indicators used to interpret the participants’ natural language.

***Insight/Pattern Recognition*** is a cognitive factor that allows individuals to identify meaningful behaviors to meet specific needs. When individuals develop insights about their practices, it will enable them to make conscious decisions about whether to disrupt this pattern, substitute a healthier one, or begin a new one. Tracking/self-monitoring tools and resources assist people in this process and keep their intended behavior goals at the forefront. As appropriate, the NHC introduced clients to tracking tools, guided them to pattern recognition, and prompted them to develop insights about how they hinder or support personal health goals [7,8,9]. Examples of language that indicate this construct are: expressing the discovery of a sticking point, discovering new goals and desired behaviors, re-ordering one’s priorities, or being honest with oneself.

***Self-efficacy and Independent Agency*** represent a person’s confidence to execute behaviors necessary for goal attainment (self-efficacy) and the belief that one’s actions will significantly impact health or quality of life (agency). NHC addresses low self-efficacy and agency by using strategies to strengthen them, such as eliciting personal stories (narratives) with affirming reflections on life’s accomplishments. Self-efficacy and independent agency are essential to successful behavior change and clinical outcomes [10,11,12]. Examples of language that indicate self-efficacy include expressing confidence in achieving a new goal, overcoming a known barrier to success, and believing one controls one’s destiny or future.

***Building Toward Sustainability*** is a cognitive–behavioral construct that addresses the “why” of the behavior change and the emerging core identity as one that enacts these healthy behaviors. Behavior-change literature has shown the sustainability of outcomes in the short term, such as a few weeks or months; however, sustaining results beyond six months is difficult, especially in obesity management [13]. To effect long-term sustainable changes, the NHC uses strategies to support permanent habits that help the target behavior [14,15], such as a “coupling ritual” anchoring medication to eating breakfast at the kitchen table every morning, which then transfers (i.e., cascade effect) to including healthier food choices at breakfast [9,13,14,15,16,17,18,19,20]. NHC facilitates the cascade effect through guiding insight and pattern recognition. The stability of these linked behaviors supports individuals to think of themselves differently. Examples of language that indicate sustainability include forecasting healthier activities in the future, developing a plan, and imagining oneself as a “healthier self”.

***Resiliency*** is “the process of adapting well in the face of adversity, trauma, tragedy, threats, or significant sources of stress—such as family and relationship problems, serious health problems, or workplace and financial stressors” [18]. Furthermore, resiliency is the ability to manage the ups and downs of everyday life with attention, strength, practice, and hope. Resiliency is multifaceted, including thinking, reflecting, trying, forging, learning, and recommitting. It also includes applying strength in one area of accomplishment to another area of the desired action [19,20,21,22]. The NHC encourages resiliency using affirmations, reframing, and cognitive restructuring strategies. Examples of language that indicate resiliency are problem solving for overcoming barriers and re-committing to a goal after trial and error.

***Change talk*** is considered a key treatment mediator for the motivational interviewing (MI) approach, with a solid research base regarding its positive influence on behavior change and health outcomes [23,24]. Change talk (C-Talk) is the language people use in their natural conversations to verbalize the need, desire, or intention to change from the status quo. Multiple studies have shown that C-Talk predicts favorable outcomes, which expresses commitment to change, activation, and taking steps toward change [25,26,27]. NHC uses MI to facilitate the progression of C-Talk through softening sustain talk (barriers and challenges to the targeted behavior change/goal) and strategically evoking and responding to C-Talk, an advanced MI skill set [28]. Examples of language that indicate change talk are expressing a desire to change, expressing a reason to change, and verbalizing action plans to change.

## 2. Design and Methods

Design. We employed a single group, repeated measures (1 × 3) design to determine the change in cognitive/behavioral outcomes over time, with NHC as the independent intervention variable. We hypothesize that NHC will significantly improve the specified cognitive/behavioral outcomes, change talk, resiliency, self-efficacy/independent agency, insight and pattern recognition, and building towards sustainability from baseline (time 1) to the treatment endpoint (time 3). 

Nurse Health-Coaching Methods and Strategies. All NHCs were registered nurses (RN) with at least ten years of practice experience and a minimum of a baccalaureate degree in nursing. All completed training in our NHC methods [1,4]. The program, delivered virtually using two-way video connectivity, focused on person-centered engagement, client empowerment, and cognitive–behavioral and narrative coaching—the overall communication approach based on motivational interviewing (MI) [27,28]. As previously reported [4], strategies are a “tool kit” tailored to foster positive change in the cognitive outcomes, addressing each participant’s unique contextual driving and restraining forces to health-related behavior change [29,30,31,32] (see Figure 1).

The primary premise of these analyses is that, if the NHC strategies were used as indicated in the framework, i.e., addressing inhibiting and driving factors, including social determinants of health (SDH), to achieve a positive change in the cognitive behavioral outcomes, then (1) there will be a progressive improvement in cognitive/behavioral indicators measured at baseline, at the midpoint of the intervention, and the end of the NHC 8-week intervention, (2) there will be no difference in the outcome measurement of a subgroup of participants with SDH from that of the whole sample.

Negative social determinants of health were defined for our sample as primarily meager income (<USD 20,000/year) and early-onset hypertension, a well-recognized health disparity. In addition, low income is associated with significant resource barriers influencing one’s neighborhood, access to groceries, and transportation. These socio-economic and structural barriers increase health disparities [33,34,35,36]. Additionally, African American and unemployed were additional characteristics of the twelve participants we characterized as having negative social determinants. Anecdotally, these participants reported living in neighborhoods characterized as “unsafe” or with limited environmental resources, such as public transportation and grocery store access.

Development of the Measurement Tool for Coding Cognitive/Behavioral *Indicators of Health Behavior Change* (IHBC). The indicators of each cognitive/behavioral measure described above were developed, and face/content validity was established with independent experts. The experts were nursing/behaviorist faculty members at the university who were not involved in the development of the indicators. We developed a measurement tool for coding that was designed to allow trained health coach experts (called “coders”) to assess the presence and frequency of the indicators in the natural language content of audio segments of participants in the intervention arm of the RCT. For the change talk component, scoring was derived from two previous studies/methodologies [30,31], resulting in a total score that considers both the quantity and strength of the client change talk, with a possible score range between 0 and 10. For the other cognitive indicators, namely resiliency, self-efficacy/independent agency, insight and pattern recognition, and building towards sustainability, the score was awarded 0–5 based on the strength of the client talk representing that factor, with a 0 designated if there were no answers that supported the measure, and with 5 assigned if there was significant evidence of reports that helped the effort. For these four cognitive factors, descriptors of possible types of expressions under each area guided the coder. The coder provided examples of client answers supporting their score of all measures. 

Outcome Measures. The cognitive/behavioral outcomes are change talk, resiliency, self-efficacy/independent agency, insight and pattern recognition, and building towards sustainability, as measured by the IHBC tool described above. 

Audio Session Sampling Method. We used a two-step method for randomly selecting the 20 min audio-recorded session that was analyzed at each time point. In step one, sessions were randomly selected that represented time progression during the 8-week program: thirty-minute samples were drawn at three different points in the intervention: early (weeks 2–3), mid (weeks 4–5), and late (weeks 6–8). Then, in step two, we randomly selected a 20 min segment to analyze from the 30 min segments selected in step one.

Establishing Inter-Rater Reliability of Session Coding. Two coders were used in 10% of coding sessions. One of two gold-standard coders was used as the second coder such that gold-standard coders #1 and #2 each double coded 5% of the sample. Gold-standard coders are the two principal developers of the measurement tool: co-authors M.F.R. and S.W.B. We used the online system from www.random.org to randomly select 10% of sessions from each coder. After double coding, we determined that the ICC ranged from 0.876 to 0.922, with an average ICC of 0.89 across all constructs.

Testing Construct Validity of the Instrument. We examined the construct validity of our instrument using exploratory factor analysis at each of the three measurement times. We used an orthogonal rotation. Factors were extracted based on the variance explained and scree plots. Factor loadings less than 0.30 were not considered substantial and were suppressed for ease of interpretation.

Testing Change in Cognitive Verbalization Over Time. We performed a repeated-measures ANOVA with a fixed effect of time (beginning, middle, end) to test the change in cognitive verbalization over time. The parameter of interest was the main effect of time. Pairwise comparisons from baseline and *p*-values are based on Tukey’s post hoc HSD test. Standard checks for violation of assumptions were conducted. To explore the effect of SDH, we added a covariate indicating group membership.

## 3. Results

Sample. Fifty-nine participants were randomized to the NHC intervention arm of the RCT. Twelve participants were identified as having negative social determinants of health. The demographic characteristics of the randomized sample (*N* = 59), the SHD participants (*N* = 12), and the non-SDH participants (*N* = 47) are described in Table 1. Fifty-six participants had high-quality audio recordings of the NHC sessions after voice-distortion de-identification procedures were performed and are included in the analyses (11 SDH, 45 non-SDH). We performed an additional comparison analysis of outcomes between the SDH group and the non-SDH group considering the SDH as an essential component of our NHC Framework (Figure 1). 

Outcomes. Table 2 displays the mean and standard deviation (SD) of the outcome measures at each time point (at weeks 2–3, beginning; at weeks 4–6, middle; at weeks 7–8, end) of the 8-week NHC program. Table 3 displays the correlation matrix of the outcome measures. Change talk is only minimally correlated with the other four variables, while the other four are moderately correlated. Factor loadings from the factor analysis are displayed in Table 4. Both the explained variance and the scree plot suggest a two-factor solution. Change talk was a distinct factor at all three points of time, independent of the other concepts. In addition, the remaining constructs all loaded on the other factor.

Table 5 displays the mean differences from the baseline. Negative differences indicate that the values increased with time. All the differences except early insight were statistically significant and increased with time. The outcome of sustainability violated the ANOVA assumptions and was not modeled. The model for the composite latent variable (the mean of the four variables, including sustainability) satisfied the distributional assumptions and showed an increase over time.

SDH group analysis. Table 6 shows the means (SD) of the latent cognitive variable for the SDH and non-SDH groups. Combining both groups, the main effect for a time showed significant improvement over time at *p* < 0.0001. There was no group interaction effect *p* = 0.66. However, the a priori power of this test was low. 

## 4. Discussion

Other than the classic MI constructs, no other published studies have assessed evidence of multiple cognitive/behavioral factors theoretically associated with behavior change from the natural language speech of clients throughout a coaching intervention. An exception is an adaptation of the Motivational Interviewing Skill Code (MISC 2.5) [25,26] for heavy drinkers (MISC-SE) that includes the MI constructs of change talk and sustain talk, with additional self-exploratory measures associated with heavy drinking and goal setting [36]. Our measurement tool consists of a score for client change talk, which we showed to be distinct from other measured cognitive/behavioral outcomes.

Our analyses demonstrated significant improvement in change talk and the other four factors over time, aligned with the temporal progression in the NHC intervention. Our factor analyses indicated two distinct factors at each measurement point of the study—at the baseline, at the mid-point, and the end of the intervention—demonstrating the stability of the factor constructs over time. A composite latent variable encompassing the domains of resiliency, insight/pattern recognition, sustainability, and self-efficacy/independent agency was shown as distinct from change talk, which is classically associated with health-behavior coaching [32]. Our newly developed measurement tool, IHBC, proved to be stable in structure over time and sensitive to change, aligned with the progression of the NHC intervention. 

We recognize that the high correlation among the four cognitive factors of insight, resilience, sustainability, and self-efficacy/independent agency may be an empirical artifact of how the indicators were described and measured, failing to demonstrate sufficient distinction when an actual difference exists among them. A greater understanding of the cognitive indicators of behavior change will help elucidate distinctiveness and associations responsive to various coaching strategies and methods. An important consideration is the design of the coaching method focused on facilitating change in both the quantity and quality of specific cognitive/behavioral factors. It should be noted that our “sustainability” indicators did not hold up well to the assumptions of our statistical procedures, suggesting that the indicators identified may not represent the construct of interest. A recent review shows that long-term sustainability beyond a few months has not yet been observed in most coaching or similar interventions across populations [33]. This suggests that the predicates of sustainable behavior change are not yet known and demand more exploration. 

Moreover, an important finding is that our analyses also showed no difference in outcomes for a small subgroup of participants with negative SDH compared to the whole sample’s results. While many interventions strive to reduce health disparity and negative SDH, little research has been published regarding successful lifestyle interventions despite negative SDH [34]. Addressing the facilitating and inhibiting factors associated with SDH is a distinct feature of our NHC framework and training. NHC was designed to address issues related to negative SDH using the strategies to mitigate common inhibiting internal factors such as low self-esteem and external factors such as limited income. Specifically, low self-esteem is addressed by showing consistent unconditional positive regard, positive affirmations noticing every positive attribute or behavior, recognition of resiliency, and eliciting the participants’ recognition and insight into these positive aspects. Regarding environmental factors, for example, all of our negative SDH participants were aware of and used all publicly available resources, such as the Supplemental Nutrition Assistance Program (SNAP) or food banks. Transportation was more difficult to access, but each participant had an alternative, including collaboration with family members or neighbors. This competency in using resources and alternative planning is an indicator of resiliency, a positive attribute that the NHCs reinforce and build upon during the coaching sessions. Additionally, it has been established that there is much inherent bias in healthcare by practitioners towards underserved and poor populations, which can manifest in a lack of belief in their client’s ability and inhibition of efforts to engender empowering strategies to address complex lifestyle-related changes with these clients [35]. While the SDH analysis is underpowered to detect group effects, the trends indicate that both groups improved similarly over time, suggesting that the NHC benefits individuals with and without negative SDH. This relevant outcome warrants more study with rigorous design to confirm.

Our research outcomes are encouraging, as we observed an increase over time for client change talk and multiple cognitive factors that are highly correlated. Indicators of positive cognitive change recognized in the natural language of clients during the coaching process are important guideposts in managing the application of strategies to foster continued growth and progress in clients. In addition, the study results indicate that our NHC training program, designed to address these targeted outcomes, was successful. 

Our newly developed IHBC measurement tool for coding natural language responses during coaching sessions is linked with clinician training intended to address multiple higher levels of cognitive–behavioral constructs that impact numerous health behaviors and is commensurate with the best practice of incorporating various health-behavior change interventions to maximize synergy and cost-effectiveness. Short-term interventions such as our NHC 8-week intensive program are optimized when there are clear guideposts of client progress and for applying coaching strategies for further growth [36]. Future research efforts will be directed at further development and validation of the IHBC tool and identification of the link between specific NHC strategies and cognitive/behavior outcomes.

## 5. Limitations

This study was conducted in an older adult population with chronic conditions unique to the individuals sampled. While our NHC model and application methods are theoretically appropriate for adults with a range of chronic conditions influenced by lifestyle factors, the results of this study cannot be generalized to others. The specific dominant disease of our sample is hypertension, and our results are related to this condition. 

## 6. Conclusions

A short-term, eight-week NHC program utilizing a standard set of intervention strategies shows promise in improving cognitive–behavioral indicators associated with health-behavior change. The IHBC measurement device for coding natural language responses is a promising method for assessing “real-time” evidence of these indicators in the client speech elicited during coaching sessions. When linked to specific coaching strategies, empirical evidence from the natural language of clients has the potential to develop, test, and mature interventions for NHC for a myriad of resistance factors.

## Figures and Tables

**Figure 1 ijerph-20-00416-f001:**
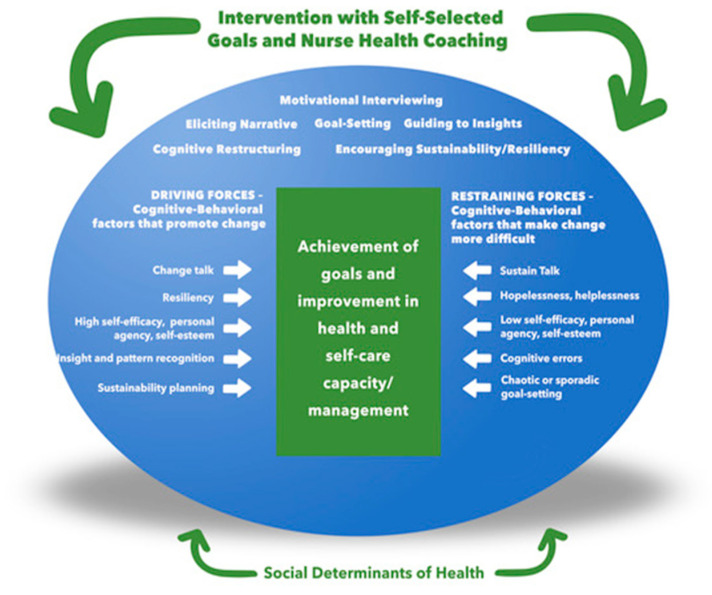
The Nurse Health Coaching (NHC) Framework *. * Adapted from Kurt Lewin’s Force Field Analysis Model.

**Table 1 ijerph-20-00416-t001:** Demographic characteristics of the whole sample (*N* = 59) and the 11 participants with negative social determinants of health.

	Total	SDH	Comparison (Total-SDH)
	*N* = 59	*N* = 12	*N* = 47
Sex			
Female, *n* (%)	48 (81.4)	12 (100)	36 (76.6)
Male, *n* (%)	11 (18.6)	0 (0)	11 (23.4)
Prefer not to answer, *n* (%)	0 (0)	0 (0)	0 (0)
Education			
Less than Bachelor’s, *n* (%)	23 (39)	8 (66.7)	15 (31.9)
Bachelor’s, *n* (%)	17 (28.8)	1 (8.3)	16 (34)
Graduate/Professional, *n* (%)	19 (32.2)	3 (25)	16 (34)
Race			
African American, *n* (%)	29 (49.2)	8 (66.7)	21 (44.7)
White, *n* (%)	26 (44.1)	4 (33.3)	22 (46.8)
Other, *n* (%)	4 (6.8)	0 (0)	4 (8.5)
Prefer not to answer, *n* (%)	0 (0)	0 (0)	0 (0)
Income group (USD)			
0–20,000, *n* (%)	12 (20.3)	12 (100)	0 (0)
21,000–50,000, *n* (%)	19 (32.2)	0 (0)	19 (40.4)
51,000–100,000, *n* (%)	24 (40.7)	0 (0)	24 (51.1)
Greater than 100,000, *n* (%)	3 (5.1)	0 (0)	3 (6.4)
Dominant condition/disease	Hypertension	Hypertension	Hypertension

**Table 2 ijerph-20-00416-t002:** Trends of the Outcome Measures Over 3 Time Points (mean (SD)).

	Beginning (*N* = 57)		Middle (*N* = 55)		End (*N* = 55)	
Variable	Mean	Std Dev	Mean	Std Dev	Mean	Std Dev
Change talk	3.48	1.54	4.89	1.96	6.35	2.48
Resiliency	3.32	0.37	3.61	0.48	3.95	0.57
Sustainability	3.22	0.35	3.39	0.48	3.78	0.62
Insight/patterns	3.35	0.37	3.51	0.49	3.85	0.58
Self-efficacy/agency	3.25	0.44	3.60	0.48	3.98	0.52

**Table 3 ijerph-20-00416-t003:** Correlation Matrix of Measures.

Kendall Tau b Correlation Coefficients, *N* = 167					
Prob > |tau| under H0: Tau = 0					
	Change Talk	Resiliency	Sustainability	Insight	Self-efficacy
Change talk	1	0.33547	0.30563	0.24582	0.35116
		<0.0001	<0.0001	<0.0001	<0.0001
Resiliency		1	0.58581	0.46473	0.60005
			<0.0001	<0.0001	<0.0001
Sustainability			1	0.5756	0.66914
				<0.0001	<0.0001
Insight				1	0.53515
					<0.0001
Self-efficacy					1

**Table 4 ijerph-20-00416-t004:** Factor Analysis of Variables at Each of Three Time Points.

**Time 1**			
**Rotated Factor Pattern**			
		**Factor1**	**Factor2**
Change talk	Change talk	0.22171	0.84727
Resiliency	Resiliency	0.70734	0.24396
Building for sustainability	Building for sustainability	0.77762	0.27138
Insight, pattern recognition	Insight, pattern recognition	0.59524	0.07602
Self-efficacy	Self-efficacy	0.76049	0.26224
**Time 2**			
**Rotated Factor Pattern**			
		**Factor1**	**Factor2**
Change talk	Change talk	0.08402	0.98427
Resiliency	Resiliency	0.67671	0.16138
Building for sustainability	Building for sustainability	0.90507	0.02195
Insight, pattern recognition	Insight, pattern recognition	0.64767	0.03519
Self-efficacy	Self-efficacy	0.78087	0.05486
**Time 3**			
**Rotated Factor Pattern**			
		**Factor1**	**Factor2**
Change talk	Change talk	0.03561	0.99836
Resiliency	Resiliency	0.76151	0.06173
Building for sustainability	Building for sustainability	0.87490	0.02052
Insight, pattern recognition	Insight, pattern recognition	0.85054	0.00836
Self-efficacy	Self-efficacy	0.84810	0.02554

**Table 5 ijerph-20-00416-t005:** Analysis of variance (ANOVA) with a fixed effect of time (beginning, middle, end) *.

Variable	Mean Difference (CL) Beginning–Middle	*p*-Value	Mean Difference (CL) Beginning–End	*p*-Value
Change talk	–1.4 (–2.3, –0.5)	0.0009	–2.87 (–3.78, –1.96)	<0.0001
Resiliency	–0.28 (–0.5, –0.07)	0.0058	–0.62 (–0.84, –0.41)	<0.0001
Insight	–0.16 (–0.38, 0.06)	0.1993	–0.5 (–0.71, –0.28)	<0.0001
Self-efficacy	–0.35 (–0.56, –0.13)	0.0006	–0.73 (–0.94, –0.51)	<0.0001
Latent variable	–0.24 (–0.42, –0.06)	0.0066	–0.6 (–0.79, –0.42)	<0.0001

* Pairwise comparisons and *p*-values are based on Tukey’s post hoc HSD test. The following table shows the mean differences from the baseline.

**Table 6 ijerph-20-00416-t006:** Means (SD) of Latent Variable for SDH Group vs. Non-SDH Group *.

Analysis Variable: Latent Variable							
SDH	time	N Obs	N	Mean	Std Dev	Minimum	Maximum
0	1	45	45	3.2777778	0.3359274	2.6250000	4.0000000
	2	43	43	3.5465116	0.4261044	3.0000000	4.6250000
	3	43	43	3.9156977	0.5494322	3.1250000	5.0000000
1	1	11	11	3.3068182	0.2118640	3.1250000	3.7500000
	2	11	11	3.4431818	0.2702062	3.0000000	4.0000000
	3	11	11	3.7727273	0.3000947	3.2500000	4.2500000

* Regression analysis showed a significantly improved trend over time, *p* < 0.0001. There was no group effect (*p* = 0.66).

## Data Availability

De-identified data from this study will be available at the CT.gov registration site, record NCT05070923, https://clinicaltrials.gov/ct2/show/NCT05070923?id=NCT05070923&draw=2&rank=1 (accessed on 12 December 2022).

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
