# Peer review of "The Impact of Nurse Health-Coaching Strategies on Cognitive—Behavioral Outcomes in Older Adults"

_ijerph, 2022, doi:10.3390/ijerph20010416_

Round 1

Reviewer 1 Report

Well-written manuscript by thought-leaders in the nurse coaching field. I have no significant concerns. I do think the authors should include a short limitations section and correct a typo in table 5 (cross-out). The authors can decide if hypertension as the major diagnosis limits the generalizability of the findings. For example, if the major diagnosis were highly symptomatic (e.g., Parkinsons, rheumatoid arthritis), would the new instrument perform equally well?

Author Response

I do think the authors should include a short limitations section and correct a typo in table 5 (cross-out). 

Response: We have added a limitation section to the paper.

The cross out was removed.  Sustainability was dropped from the analysis and the reason explained in the paragraph above the table.

The authors can decide if hypertension as the major diagnosis limits the generalizability of the findings. For example, if the major diagnosis were highly symptomatic (e.g., Parkinsons, rheumatoid arthritis), would the new instrument perform equally well? 

Response: We agree with this.

Reviewer 2 Report

The paper presents the analysis of the audio-recorded natural language of participants during NHC sessions to evaluate improvement over time in cognitive-behavioral outcomes. 

Results demonstrate that this NHC program shows promise in improving cognitive-behavioral indicators associated with health behavior change in both non-SDH and SDH individuals.

Several suggestions for improving the manuscript:

1/ Introduction seems like repeating the abstract. It may be better to incorporate it with the background section.

2/ Background is confusing and could be better organized. Illustration 1 is essential but needs to be better explained, especially regarding the links among the driving forces you describe at the bottom of this illustration. 

3/ Lines 117-123 need to be clarified. I suggest rephrasing them to include the research goals or hypothesis directly and simply.

4/ Please explain what RN is in line 107. 

5/ Paragraph titled "Nurse Health Coaching Methods and Strategies." should be placed in the method section.

6/ Describe the "independent experts" you mentioned in line 131.

7/ Please add a paragraph describing the limitations of the study.

Thank you. 

Author Response

1/ Introduction seems like repeating the abstract. It may be better to incorporate it with the background section. 

Revisions were made incorporating introduction into background.

2/ Background is confusing and could be better organized. Illustration 1 is essential but needs to be better explained, especially regarding the links among the driving forces you describe at the bottom of this illustration. 

Revisions were made for clarity.

3/ Lines 117-123 need to be clarified. I suggest rephrasing them to include the research goals or hypothesis directly and simply.

We added lines 123-128 to specifically identify our design and hypotheses.

4/ Please explain what RN is in line 107

Done

5/ Paragraph titled "Nurse Health Coaching Methods and Strategies." should be placed in the method section. 

Done

6/ Describe the "independent experts" you mentioned in line 131.

This is explained in lines 153-154.

7/ Please add a paragraph describing the limitations of the study. 

A limitations paragraph has been added.